# Applications and Advances of Multicellular Tumor Spheroids: Challenges in Their Development and Analysis

**DOI:** 10.3390/ijms24086949

**Published:** 2023-04-08

**Authors:** Achilleas G. Mitrakas, Avgi Tsolou, Stylianos Didaskalou, Lito Karkaletsou, Christos Efstathiou, Evgenios Eftalitsidis, Konstantinos Marmanis, Maria Koffa

**Affiliations:** Cell Biology Lab, Department of Molecular Biology and Genetics, Democritus University of Thrace, 68100 Alexandroupolis, Greece; amitrak@med.duth.gr (A.G.M.);

**Keywords:** 3D cell cultures, multicellular tumor spheroids (MCTS), cancer, microscopy, SPIM

## Abstract

Biomedical research requires both in vitro and in vivo studies in order to explore disease processes or drug interactions. Foundational investigations have been performed at the cellular level using two-dimensional cultures as the gold-standard method since the early 20th century. However, three-dimensional (3D) cultures have emerged as a new tool for tissue modeling over the last few years, bridging the gap between in vitro and animal model studies. Cancer has been a worldwide challenge for the biomedical community due to its high morbidity and mortality rates. Various methods have been developed to produce multicellular tumor spheroids (MCTSs), including scaffold-free and scaffold-based structures, which usually depend on the demands of the cells used and the related biological question. MCTSs are increasingly utilized in studies involving cancer cell metabolism and cell cycle defects. These studies produce massive amounts of data, which demand elaborate and complex tools for thorough analysis. In this review, we discuss the advantages and disadvantages of several up-to-date methods used to construct MCTSs. In addition, we also present advanced methods for analyzing MCTS features. As MCTSs more closely mimic the in vivo tumor environment, compared to 2D monolayers, they can evolve to be an appealing model for in vitro tumor biology studies.

## 1. Introduction

Since the early 20th century, two-dimensional (2D) cell cultures have been the gold standard method for cell research [1]. Despite its success and its major contribution to biological studies, 2D cell cultures do not appropriately simulate the structure, organization, and drug resistance of solid tumors [2]. Cells growing in three-dimensional (3D) culturing systems more closely mimic the in vivo tumor environment when compared to 2D monolayers. Cell aggregates mirror the molecular signaling occurring among cells of the same tissue by enabling cell–cell and cell–matrix interactions [3]. Further, 3D cultures allow for more reliable studies concerning biological aspects such as viability, proliferation, morphology, differentiation, and drug metabolism [4].

Consequently, 3D models are a significant tool for researchers, as they lie between 2D culture systems and in vivo organismal models. The employment of 3D cell culture models, such as multicellular tumor spheroids (MCTSs), has significantly increased in recent years. MCTSs originate either from a single cancer cell lines or upon a co-culturing of cancer cells with fibroblasts, endothelial cells, or immune cells [5]. MCTS cell clusters form either by forced or self-assembly under scaffold-free conditions or through incorporating certain 3D scaffolds. The surveillance of tumor-like hypoxic conditions, as well as the subsequent release of angiogenesis-related components, can also be studied [6]. MCTSs have become an appealing model for in vitro tumor biology studies [7,8,9,10], providing an intermediate pre-clinical step for the development of anticancer drugs and new therapeutic approaches.

In an ideally shaped MCTS, the distance of every surface point from the center is the same; this symmetry provides a great model for the physiological concentration gradient of the oxygen, nutrients, soluble signals, and metabolites in tumors [11]. As a result of the nutrient and oxygen gradient, different zones are defined inside the MCTS, accompanied by differential cellular behavior (Figure 1). In the necrotic zone, at the core of the structure, the majority of cells are dead, due to hypoxic conditions, and there is a lack of nutrients and growth factors. More viable, but quiescent, cells can be found moving outwards to the surface. In the outer proliferating zone, cells are increasingly exposed to sufficient amounts of nutrients, oxygen, and growth factors [12,13]. Hence, cells near the surface proliferate faster, while the growth rate in the internal layers is reduced.

More complex 3D model systems, such as organoids or tumoroids of substantial size, have been formed using a mammary-originated hydrogel with unique features. These systems allow for further experimentation on extracellular matrix (ECM) and epithelial or tumor cell interactions [14]. In addition, biomimetic tumoroids illustrate heterogeneity and substantial angiogenesis, and are differentially affected by matrix density and stromal composition. Thus, such 3D tumor models can serve as a better prediction tool for cancer progression and angiogenesis studies [15].

More recently, innovative patient-derived organoids (PDOs) have further advanced studies on drug screening and development, as well as biomarker identification. Relevant experimental data on PDOs have already suggested that they can mimic an organism’s reaction to a drug, as they simulate a patient’s tumor features [16,17,18]. Patient-derived cancer cells and normal cells supplemented with matrix proteins result in tumoroids that are functional surrogates of native tumors, providing a translational tool for studying cell–matrix interactions, drug development, and precision medicine approaches [17,18].

In this review, we focus on MCTS formation techniques, their applications in cancer biology, along with the challenges emerging from imaging and data analyses.

## 2. Methods of MCTS Formation

The reproducible formation of MCTSs still entail technical challenges: not all cancer cells form spheroids with regular morphologies, as their handling often induces their disintegration.

In order to mimic growth conditions and the tumor microenvironment, various methods have been developed to “reconstitute” 3D tumor models. These methods are divided into two main categories, the scaffold-free and scaffold-based culture systems [19,20].

### 2.1. Scaffold-Free Systems

#### 2.1.1. Liquid Overlay Technique

The liquid overlay is the simplest technique to produce 3D cell cultures, based on the principle of “forcing” cells to aggregate or self-assemble [2,21,22,23], as they cannot attach to the underlying surface. This occurs due to artificial vessels coating with non-adhesive materials, such as agar, agarose, Matrigel, 0.5% poly-HEMA (hydroxyethyl methacrylate), and, more recently, hyaluronic acid [8,24,25,26]. As a result, cell–cell contact is promoted. The cells initially aggregate loosely, and then, eventually, spheres are formed (see Figure 2a).

This scaffold-free assay allows the formation of MCTSs of different sizes, starting from a single cell or variable cell numbers, in a cost-effective and easy way. They can form either from a single-cell type or as co-cultures [2,27,28], while single-cell-originated MCTSs can be used for medium-throughput experiments [29].

#### 2.1.2. Hanging Drop Assay

The hanging drop method is another scaffold-free approach for MCTS formation, initially applied for embryonic stem cell cultures by Johannes Holtfreter in 1944 [12]. In principle, the single cell suspension of trypsynized cells is diluted at the appropriate density and the precise volume of the suspension is dispensed on a sterile tray (which is inexpensive and can be used as a culture dish lid). Upon inversion of the tray, cells aggregate and proliferate, and spheroids are eventually formed as droplets due to surface tension and gravity [30] (see Figure 2b).

MCTS sizes can be regulated by cell suspension density, and the variation in size in replicates could be as little as 10 to 15% [31,32]. Advanced versions of the hanging drop assay involve the use of bioassay dishes in order to obtain more controllable and reproducible culture conditions, or the use of 384-well plates that are adapted to high-throughput screening devices [33].

Similar to the liquid overlay, the hanging drop method is inexpensive and does not require specialized instruments, leading to controlled sized 3D structures. However, it is more labor-consuming, and the limited liquid volume used results in restricted size and flexibility.

#### 2.1.3. Agitation-Based Methods

Another approach to develop MCTSs involves the use of agitating bioreactors; either spinner flasks [4,28,34] or rotational culture systems [35,36]. Their common underlying principle is the continuous motion of the cell suspension, either by stirring (spinner flasks) or by rotation on a horizontal axis (rotational culture systems) (see Figure 2c). As a result of the continuous motion of the suspension liquid, the cells do not attach on the substrate surface, but rather aggregate and self-assemble [33]. Agitation-based methods are used for large-scale spheroid formation. Their design allows culture nutrient enhancement, waste disposal, and the homogeneity of the culture medium. Drawbacks of these methods include the use of expensive instruments, the utilization of large quantities of culture media, the production of a heterogeneous population of MCTSs, and the mechanical damage of cells [28]. The latter problem can be surpassed in the case of rotation wall vessels (RWV) developed by NASA to simulate microgravity, and which employ low sheer force on cells while in culture [37]. Table 1 below summarizes the benefits and drawbacks of the scaffold-free methods.

### 2.2. Scaffold-Based Systems

Scaffolds are usually biopolymers with a composition that mimics the structure and mechanochemical properties of the extracellular matrix (ECM). They form porous structures that allow for the transportation of nutrients, oxygen, and waste in and out of the cell aggregates, shaping the aforementioned proliferation gradient. Scaffold properties may allow cell mobility, cell adherence to scaffold components, and can assist with the building of spheroid-initiating cores, with the potential to grow into fully developed MCTSs [38,39]. Scaffolds can be utilized in the majority of 3D culture techniques. Depending on the requirements of the cells and/or the downstream applications of the 3D cultures, scaffolds can be either used as culture medium supplements or matrices, providing structural support and microenvironmental conditions that are similar to those of the ECM.

Different tumor cell types have distinct demands from the tumor microenvironment. Therefore, various kinds of scaffolds have been designed in order to meet each cell type’s special needs. According to the support they offer, scaffolds can be divided into two main categories: hard polymeric materials, which form sponge-like or fibrous structures; and hydrogels, which form networks extensively swollen with water. Scaffolds can also be categorized into natural polymers, synthetic materials, or hybrid systems [40]. Table 2 summarizes common scaffold-based support systems along with the properties that make them useful as 3D cell culture scaffolds (also in [6]).

Polymers of animal origin show a greater similarity with the ECM, although their usage is linked to some drawbacks, such as great variability [41], as well as environmental [42] and ethical concerns [43]. In particular, collagen is one of the most common materials used for cancer cell studies. The main advantage of collagen is its cytocompatibility, while cells directly adhere without any modification or component addition. Some of its disadvantages include short-term stability, low stiffness, and high variability among different batches [44]; however, collagen and hydrogels containing collagen still remain a great option for scaffold-based systems. For example, Matrigel contains collagen type IV, laminin, fibronectin, entactin, and growth factors, and it is a widely used collagen-based hydrogel for angiogenesis, cancer development, and cancer cell migration studies [45].

Fibrin, another polymer used as a scaffold, simulates the natural matrix and is employed for wound healing, angiogenesis, and hemostasis studies [46,47]. However, fibrin is not suitable for long-term cancer cell cultures since the polymer is degraded by proteases [48].

Many polysaccharides are also used as components of hydrogels for MCTS formation. Cellulose, the most abundant polymer in plants and algae, is used to form hydrogels that possess high cytocompatibility and do not require the addition of growth factors. These features, combined with its high mechanical properties and sustainability, allow for multiple applications [49]. Starch hydrogels present biocompatibility and biodegradation, and are used in several cell-based assays [50]; meanwhile, the mix of starch and gelatin in different ratios can be used in cancer cell studies [51]. In addition, agarose, a slow degradable polysaccharide shows a high adaptiveness to cell culture. Agarose hydrogels are permeable to water, oxygen, and nutrients, leading to high rates of cell survival and growth. However, cell adhesion is limited [52,53].

Glycogen hydrogels exhibit pores and hydrophilicity while they are non-toxic for many human cell types. They have been applied in prostate cancer and osteoblast-related studies [54,55]. Lastly, hyaluronic acid (HA), a neutral polysaccharide that is abundant in the ECM, plays crucial role in cell motility, matrix organization, angiogenesis, and wound healing [56]. It is used for a variety of experimental approaches as it is a perfect scaffold for human fibroblast, hepatocyte, and keratinocyte cultures, or for studies regarding neural and vessel growth and development [57].

Polypeptide-based hydrogels are an interesting approach for cell culture applications, as they present substantial interactions with cells while they can be conjugated with other materials. However, the commercially available polypeptide-based hydrogels are expensive for large scale experiments [58,59].

Except for biopolymers, other materials have also been used as scaffolds for 3D cultures. Polyacrylamide, a synthetic polymer, used in hydrogel formation, allows experimentation on cell behavior in various microenvironments, with the change in simple parameters affecting stiffness and adhesion ability [60]. Han and colleagues developed an aqueous two-phase system comprising aqueous solutions of polyethylene glycol (PEG) and dextran (DEX). This system works under similar principles to the hanging drop method and allows MCTS formation of various cell lines, including very sensitive stem cells (Figure 3) [61].

Other methods, such as the utilization of magnetic beads [61], or other microspheres, [62] have been used as initiating cores for 3D cultures. In such systems, cells aggregate in the periphery of these spherical objects and start proliferating, gradually forming spheroids. Using these methods, cell aggregation has a great chance to occur; however, they fail to form the proliferation and metabolic gradient of actual tumors due to the fact that their cores are cell free. Moreover, the physiochemical properties of the materials used may have an impact on the cells, resulting in altered behavior during MCTS formation. Modifications, such as functionalization with ECM proteins [63], have made beads more cell-friendly, yet have not eliminated the limitations arising from the cell-free core regarding MCTS structure.

Since the matrix plays the role of the tumor microenvironment, it is essential that its composition is optimal for cell communication, mobility, and proliferation [64]. Leung and colleagues investigated the alteration in circularity and compactness of different cell lines in the presence of different concentrations of two scaffolds (collagen I and methylcellulose). They showed that different cell lines behave differently in the same concentration of each, or simultaneous, use of both polymers [11]. The simultaneous utilization of more than one scaffold systems has also been reported: Kojima and colleagues combined methylcellulose as a matrix together with polystyrene microspheres, which enabled the co-culture of Hep G2 and NIH3T3 cells [62].

Currently, novel approaches that utilize automatic 3D-bioprinting offer the opportunity to manufacture tissue-like structures, which perform one or more biological functions [65,66]. Such systems allow for the controlled positioning of cells, tissues, and biodegradable materials, bridging the gap between artificially engineered and native tissue [67]. As a result, it is now possible to form complex structures to investigate tissue function, bringing us one step closer to understanding the biological procedures taking place in vivo, both in healthy and pathological conditions. In addition, Mollica et al. have developed organoids/tumoroids in hydrogels originating from human or rat mammary tissue [14].

**Table 2 ijms-24-06949-t002:** Summary of the common scaffold-based support systems [40,68].

Scaffold Type	Properties
Collagen [69,70,71]	Enzymatically degradable Similar structural and mechanical properties with in vivo tissues Native cell adhesion ligands
Matrigel [45]	Complex mixture of basement membrane proteins, cytokines, and growth factors Cancer origin (Engelbreth-Holm-Swarm mouse sarcoma) Ideal for invasion, metastasis, and drug resistance studies
Fibrin [46,72,73]	Enzymatically degradable Good substrate for wound-healing studies Limited utility due to low mechanics
Cellulose [49,74,75]	Good biocompatibility Low cytotoxicity Good mechanical properties
Starch [76,77]	Generally insoluble in water or alcohol, but can be dispersed in water upon heating Viscosity increases upon cooling and forms hydrogels by physical crosslinking
Agarose [52,53]	Slow degradability Poor injectability Low cell adhesiveness High hydrophilicity
Glycogen [55,78]	High molecular weight Spherical glycogenin protein core Contains the multivalent binding sites of lectins
Hyaluronic acid (HA) [79,80,81]	Many potential chemical modifications enable considerable tunability Reactivity with cell receptors, but must be modified with adhesive ligands to permit cell attachment
Polypeptides [58,82]	Typically formed by self-assembly Protein engineering enables great design flexibility Useful for soft-tissue applications and in conjunction with other materials
Polyacrylamide (PA) [83,84]	Wide range tuning of substrate mechanics Probably the most standardized material for making hydrogels and in using for culture
Polyethylene glycol (PEG) [85]	‘Blank slate’ synthetic material enables a plethora of modifications Pre-modified versions and various molecular weights are commercially available Can be engineered to present different adhesive ligands and to degrade in a passive, proteolytic, or user-directed way

## 3. Applications of MCTSs in Anti-Cancer Studies

Cancer is a challenge for the biomedical community due to its high morbidity and mortality rates worldwide. Since several in vitro findings do not have the expected therapeutic impact in vivo, there is a great need to better resemble the in vivo conditions in in vitro models [86].

MCTSs are used as 3D cancer models, simulating cell–cell interactions and several chemical and biophysical characteristics of solid tumors. The cell–cell interactions affect the resistance of cancer cells to radiotherapy or chemotherapy. To this end, cancer cells exhibit higher survival rates post-irradiation when cultured and embedded in the ECM, which is a phenomenon called “cell adhesion-mediated radioresistance” [87], when compared to standard 2D culture. This ‘contact effect’ plays a crucial role in sensitizing cancer cells to radiotherapy [88]. Cell behavior post-irradiation in 3D co-cultured MCTS models is also altered by the presence of normal cells (fibroblasts or endothelial cells co-cultured with the cancer cells), enhancing the resistance of cancer cells to radiation [89,90].

Drug resistance in 3D MCTSs compared to 2D cultures is also shown [91] after treatment with several anticancer drugs, including 5FU (5-Fluorouracil) and gemcitabine [91]. This is possibly because many drugs cannot penetrate through the inner body of the MCTSs. Moreover, the expression levels of genes related to apoptosis, proliferation, or tumorigenesis [92], as well as levels of various miRNAs, such as miR-21 and miR-335, present different patterns in 3D when compared to 2D cultures [93]. Lazzari et al. showed increased chemoresistance to Doxorubicin in the MCTSs formed upon co-culturing cancer cells with different cell types, including fibroblasts and endothelial cells [89]. These findings confirm the important role of normal cells for cancer cell survival.

Cell migration is a fundamental process involved in cancer progression. The ability of cancer cells to migrate and invade underlying tissues serves as a marker of their metastatic potential. The vast majority of cell migration and invasion studies employ 2D cultures. However, the lack of cell-to-cell interactions and the differences in physical conditions between 2D cultures and tumors lead to ambiguous results. Moreover, molecules associated with metastasis demonstrate different expression levels in 2D and 3D cultures [94,95]. Since cell invasion can be regulated by the differential density of the extracellular matrix, MCTSs and other 3D models are more physiologically relevant systems by which to investigate the role of the ECM in cancer cell proliferation, progression, and invasion [15].

The ECM is one of the main components of the tumor microenvironment. Metabolic gradients, chemical compounds, hypoxic conditions, and cell heterogeneity constitute a group of factors that should be taken into consideration in anticancer studies. Further, 3D models include a variety of cell types for clarifying the cancer cell behavior in physical microenvironments. To this end, mixed-population 3D structures were developed, including stromal cells, such as cancer-associated fibroblasts (CAFs), and immune cells, such as monocytes. CAFs are associated with the enhancement of an inflammatory tumor microenvironment, which could contribute to anticancer drug resistance and promote cancer cell invasion. In this model, monocytes are activated in tumor-associated macrophage (TAM) phenotypes without the need of other supplements [96]. Moreover, Cuccarese et al. suggest that macrophages promote metastatic phenotypes in lung cancer due to VEGF and MMP1 factors [97].

The cytotoxic effect of different drugs or irradiation [98,99] in MCTS can be macroscopically measured by calculating their diameter, as well as by assuming the structure is a perfect sphere. Another simple but time-consuming method is to count cells after incubation with the anticancer drug, following the disassembly of the MCTSs. Monico et al. have also employed the resaruzin metabolism assay in MCTSs originating from melanoma cell lines [100]. Data on cell viability and apoptosis using MCTSs could also be obtained via confocal microscopy and flow cytometry. Although these techniques offer useful information regarding cellular responses, they are not optimal for large scale screening [101].

### 3.1. Cancer Cell Metabolism

Normal cells produce ATP mainly via the Krebs cycle and through oxidative phosphorylation. However, cancer cells utilize glycolysis to produce ATP, when under aerobic or anaerobic conditions, bypassing the Krebs cycle (Warburg effect). Glucose intake is dramatically higher in cancer cells when compared to normal cells, and glucose transporters and glycolytic enzymes are overexpressed [102]. Furthermore, cancer cell metabolism is affected both by the presence of normal cells and the tumor microenvironment. Normal cells support adjacent cancer cells via the reverse Warburg effect, constituting the glycolytic part of tumor metabolism [103,104,105,106].

Cell metabolism is a main contributor to cancer cell chemo- and radioresistance. MCTSs better mimic tumor-metabolism-related events than 2D cultures. Cancer cells may take advantage of anaerobic glycolysis as a result of the proliferation gradient in MCTSs. In addition, cancer cells in 3D cultures demonstrate an increased utilization of the pentose phosphate pathway, which is associated with NAPDH production and increased fatty acids concentration [107]. Moreover, elevated HIF-1a expression levels have been measured in the most inner layer of MCTS models due to oxygen deficiency. These results are in agreement with studies that associate low oxygen concentrations with the induction of hypoxia, promoting aggressiveness and the metastasis of cancer cells [108,109]. Moreover, hypoxic conditions change the autophagic, metabolic, and apoptotic profile of cancer cells, while mitochondrial activity and DNA damage rate may also be altered [110]. Thus, employment of 3D culture models in the study of metabolism-related pathways could provide more accurate and representative information on the actual events that take place within in vivo tumors.

### 3.2. Cell Cycle Research

Cell cycle progression depends on the position of cells within the MCTS. More specifically, the distance of a cell from the MCTS surface affects the cell cycle state, in part—at least—due to the non-homogeneous distribution of nutrients, as well as gas and cellular waste [111]. MCTSs have been used for estimating the efficiency of antitumor compounds, as these models develop multicellular resistance (MCR). The latter occurs as a result of cell–cell interaction, inefficient drug penetration, and the resistance of quiescent cells located deeper in the MCTS.

MCTS growth proceeds through three phases: initially, proliferation is similar throughout the volume of a growing multicellular tumor spheroid. Next, a proliferation gradient is formed with the center of the spheroid remaining viable, but the cell cycle is arrested. During the last phase, a necrotic core is developed. Cells within the tumor spheroid tend to pass the restriction point and commit to the cell cycle the closer they are to the surface; thus, an increase in G1 phase duration is observed in cells located closer to the center of the spheroid [112,113]. Significant differences in the S/G2 cell ratio are observed between the inner and outer layers of the MCTS, with the outer region being enriched in S/G2 cells [114] (see Figure 4).

Several cell analysis techniques, including flow cytometry, immunoblotting, and PCR, although invaluable, fail to give spatial information. The fluorescence ubiquitination cell-cycle indicator (FUCCI) has been reported to assort the different phases of the cell cycle in real time, using the fluorescent tagging of proteins expressed in specific phases of the cell cycle [113]. This way, one could monitor the novel therapy targeting of non-dividing cancer cells and improve the efficacy of cancer chemotherapy [115].

Classification of the cell cycle state, as a function of cellular localization within MCTSs, can also be performed using DNA content analysis [111]. A reduction in nucleus size has been monitored as a function of distance from the MCTS perimeter, due to differences in stress distribution, which could prove useful in drug development studies. However, the MCTS analyzed had a diameter of 120–150 μm, missing a necrotic core; therefore, larger MCTSs would possibly show different results.

The crosstalk between tumors and the components of the microenvironment, such as ECM, fibroblasts, and endothelial cells, is essential for tumor progression. The size and ratios of the different types of cells comprising the MCTS may affect cell-cycle-related phenomena. Browning et al. created a mathematical model for the study of MCTS structures as a function of overall size. The melanoma cell line’s MCTS had a limited size, independent of initial seeding cell number [116].

Desmaison et al. investigated the mechanical stress impact on MCTS growth. Confined MCTS forced to a rod shape exhibited proliferative cells throughout the whole MCTS, except from a small area in the center, which is in contrast to a control MCTS, where proliferative cells were restricted to the outermost layers. This mitotic cell accumulation in mechanically confined MCTS cells was attributed to mitotic arrest. Bipolar spindle assembly was also impaired, showing an increase in monopolar or tripolar spindles in confined MCTSs [117].

In order to define cell position in the large MCTSs, different microscopy techniques, such as multi-photon and light sheet microscopy (or selective plane illumination microscopy, i.e., SPIM) can be employed [118,119]. Both imaging approaches allow deep penetration near their core. Multi-photon microscopy supports imaging of up to 500 μm in depth, and light-sheet-based microscopy allows for numerous MCTS views and has become the most commonly used method for imaging their entire volume [120,121].

Eismann et al. have recently developed a SPIM-based screening workflow for the identification of mitotic phenotypes in MCTSs following siRNA treatment or epigenetic alterations. Depending on the gene silenced, five mitotic phenotype clusters were identified and categorized from nearly identical to control-siRNA-treated ones, as well as to substantial cellular and mitotic defects. This method for the investigation of MCTS mitotic phenotypes could prove invaluable for cell cycle research, cancer treatment optimization, as well as the screening and evaluation of new biomarkers [122].

## 4. Imaging and Analysis of MCTS

Optical microscopes are valuable tools to monitor MCTS and other 3D structures, such as organoids. Depending on the experimental stage and needs, different microscopes and imaging methods are utilized. Brightfield/phase-contrast microscopes, best equipped with a digital camera, can be used to quantify cell aggregation, spheroid formation, and can test different compounds without the need of fluorescent dyes [123,124,125]. Although, light transmitted microscopes originally capture a 2D projection of a 3D sample, by using computational methods, the 3D geometry of the spheroid can be reconstructed and 3D morphological parameters, such as volume and sphericity, can be extracted and quantified [23]. The above approach assumes some level of local symmetry around the MCTS axis; thus, it is best suited for well-established MCTS formation methods and can be used to improve reproducibility by selecting MCTSs with similar morphology.

Fluorescent microscopes, such as widefield and confocal, can be employed to monitor MCTSs. Here, two limiting factors need to be addressed: first, the penetration ability of fluorophores inside the MCTS; second, the ability of microscopes to capture images deep inside thick and non-transparent samples.

Characterization of MCTS layers can be performed using a widefield microscope [126]. The critical and most challenging step is fluorescence staining. As we and others have observed, fluorescent agents that do not require cell permeabilization, or they are compatible with live cell imaging, can penetrate deep inside a specimen (of 600 μm in diameter) within 6–12 h of incubation. Antibody-based staining is challenging and greatly depends on the sample size and geometry; therefore, protocol optimization is required [127]. To overcome difficulties arising from a larger size, MCTS can be sectioned and then stained. Slices can be mounted on coverslips and imaged in a widefield or confocal microscope. Ideally, cells could be genetically engineered to co-express the proteins of interest with a fluorescent tag [112].

Microscopes offering optical sectioning, such as confocal or light-sheet-based microscopes (LSFM), are able to capture the naive geometry of MCTSs. Confocal microscopes acquire images with high spatial resolution, but are suited to capture images of small and transparent samples due to their limited penetration depth [121]. Moreover, photobleaching and phototoxicity effects are high, as the imaging of multiple planes result in the heavy illumination of samples. On the other hand, the orthogonal arrangement of the excitation and detection arms in a light-sheet-based microscope (LSFM) can achieve higher penetration depths by using objectives with lower NA, with simultaneous high sub-cellular resolution [119,128,129]. Moreover, samples can be mounted without affecting their three-dimensional geometry. At the same time, since each plane is illuminated once, the phototoxicity and photobleaching levels are minimized when keeping the sample intact is critical or the number of fluorophores are limited [130]. High contrast images can be acquired using short exposure times, thus increasing temporal resolution [131]. Finally, superior datasets with isotropic resolution can be reconstructed via multi-view fusion, either by sample rotation or by recording multiple sample views using dual-side illumination and dual-side detection optics [132,133,134,135].

There are many implementations of light-sheet-based microscopes [131,136], but all of them fall either into the single (or selective) plane illumination microscopy (SPIM) or to the digital scanned laser light-sheet-based microscopy categories. Most LSFM systems are compatible with sample clearing methods, which can improve imaging depth and image quality by eliminating scattering and absorbing artifacts [137,138,139,140]. Some recent implementations combine long-term light-sheet-based imaging with high-throughput screenings on MCTSs with sub-cellular resolutions [122,141].

Morphological features of MCTSs from 2D images, either from brightfield/widefield microscopes or fluorescent sections, can be quantified using the open-source ImageJ [142]. Manual annotation of their periphery is sufficient to extract features, such as area, circularity, and major/minor axes length, to quantify its growth and formation, as well as to evaluate compound treatments [124]. Macros can be created where automation is required, as in time-dependent experiments. Usually, background/foreground separation can be performed by using threshold methods followed by morphological operations to fill gaps or clear small debris. In cases where threshold methods cannot separate the background from the foreground, machine-learning tools can be utilized. ImageJ plugins [143,144,145] offer user friendly interfaces to create ground-truth annotations, as well as to train and deploy pixel-based classification models.

Confocal, as well as light-sheet-based, microscopes generate terabyte scaled datasets. These datasets are challenging both for analysis and visualization, and usually require special software and hardware. Many commercial software can handle large datasets, including Imaris (by Oxford Instruments) and vision-4D (by Zeiss). Moreover, the ImageJ plugin BigDataViewer [146] is an open-source solution designed for multi-channel and multi-angle LSFM sequences (Figure 5).

Single-cell level analysis of MCTSs is crucial for understanding cellular processes, such as proliferation [120,147], migration, and invasion [148]. Segmenting cell nuclei within spheroids is usually the first step for single-cell quantitative analysis. Many classic pipelines rely on initial thresholding, followed by watershed algorithms to separate adjacent nuclei [111,149]. Alternatively, this can be achieved by machine and deep learning methods. Pixel-based classification tools are used to perform semantic segmentation and to classify each image voxel as the foreground (nuclei) or background [150]. However, depending on image quality and nuclei density, post-processing might be necessary [151]. On the other hand, deep learning methods can be used to perform instance segmentation and to label individual cells. Deep-learning-based segmentation can achieve very high levels of accuracy, but it requires time investment to create the ground-truth data for training the models. Cellpose is accompanied by a user-friendly interface used to annotate data, whereas StarDist relies on third-part annotation tools, such as Labkit or Paintera [152,153] (https://github.com/saalfeldlab/paintera, accessed on 28 November 2022).

Moreover, deep-learning models are useful for object classification. Eismann et al. used convolution neural networks (CNN), with very high accuracy, to classify the segmented nuclei in different mitotic phases. Similarly, Beghin et al. trained and applied the YOLOv5 CNN to classify cells according to their cell phase or to define them as mitotic or dead. Additionally, a CNN was used to classify the entire organoid according to its shape, and with 99% accuracy. In the past few years, deep learning (DL) methods proved to be a powerful tool for microscopy data analysis. Experts, as well as non-expert, users can access DL methods via the ZeroCostDL4Mic platform [154]. ZeroCostDL4Mic offers access to many broadly used models to perform segmentation, object detection, and denoising without the need of coding. Moreover, as model training can be performed on the cloud using Google Collab’s computational resources, there is no need for expensive hardware.

### 3D Cultures vs. In Vivo Tumors—What Is Missing

Although MCTS are designed to more accurately reproduce the gradients of the nutrients and oxygenation found in the tumor microenvironment, the added complexity of that environment can also present challenges to experimental design, as well as to monitoring and acquisition.

The culture method has to be chosen depending on the model system being investigated, the desired throughput, and the experimental goals. The experimental design needs to take into consideration whether reagents will reach the center of a larger MCTS or if there is a need for lytic assays and disrupting the cells within these structures, thus losing spatial information.

The microscopic imaging of tumor spheroids is challenging due to the light scattering nature of their 3D architecture. Live imaging of the MCTS is even more complicated, requiring compatible in vitro culture chambers attached at the microscope. Light-sheet-based fluorescence microscopes have been the method of choice to study MCTS development over longer time scales. Fluorescent imaging of fixed MCTSs is typically limited to the outermost cell layers, due to the poor penetration ability of fluorophores inside the spheroid.

Light-sheet-based microscopes also generate challenging datasets, requiring special software and hardware for both visualization and analysis. Sample clearing and employing machine and deep learning methods can be used to perform single-cell segmentation, leading to single cell analyses on MCTSs, both in time and space.

Over the last few decades, researchers have invested a lot of effort into the manufacturing of suitable systems that will allow the culturing of in-vivo-like structures. Currently, however, it has become clear that systems that entail one, or very few, easy-to-acquire components fail to mimic the tissue environment. Matrigel is the most well-known material for almost all kinds of 3D cultures. Nevertheless, its exact composition cannot be controlled since the amount of every component varies between different batches, resulting in low experimental reproducibility [155,156]. Thus, the utilization of one type of scaffold circumvents reproducibility issues. People have been using collagen and/or hyaluronic acid in their 3D culture systems due to their good biocompatibility that allows cell motility and signaling patterns [157,158]. Used alone, though, they still lack the complexity of multi-component tissue environments. Additionally, it is very common that one-component hydrogels have limited mechanical properties, which poses a major problem when culturing solid tissues. There are synthetic polymers, such as PEG, PCL, and PLA/PGA, of which their mechanical properties can be adjusted to some extent [159,160,161]. These materials, however, often contain reactive chemical groups that may cause alterations in cell physiology [162]. Moreover, after long term culturing, cytotoxic degradation products may be released [163].

Many of the aforementioned problems can theoretically be circumvented if the composition and the mechanochemical properties of the desired ECM are known. This way, one could manufacture a hydrogel that contains all protein factors needed and adjust the stiffness to the appropriate degree, such that cells behave as in a real tumor microenvironment. In many cases, however, this is not adequate in order to reproduce the events taking place in vivo. When trying to establish a tumor model, for example, there are many other parameters that have to be taken into consideration. The tumor ECM has very different composition from normal tissues [164]. Moreover, tumor microenvironments are shaped by “tumor stroma”, which mainly consists of the basement membrane, fibroblasts, immune cells, and vasculature. Each component is indispensable for the reproduction of the “in vivo situation” of a tumor, which affects the differentiation, proliferation, invasion, and metastasis of cancer cells [165].

Currently, many advanced approaches have been introduced to tissue research that allow for the combination of many of the aforementioned features. Many labs use microfluidic platforms that can mimic, to a great degree, the (patho)physiological functions of tissues and organs [166,167,168,169]. These chip-like devices consist of microfluidic channels connected to each other, such that cells from different compartments can communicate via their shared fluid. Their revolutionary advantage is that many parameters, such as mechanical forces, the components’ concentration or gradient, cell types, and/or tissue polarization can be controlled independently [170]. Additionally, due to their small size, the cost of the culture can be significantly decreased. Systems such as these, however, are prone to technical impairments, not only due to handling manipulations, but also due to the non-desired properties of the fabrication materials, resulting in decreased reproducibility and reliability [171,172,173].

Another important parameter to be taken into consideration while developing representative tumor systems is the heterogeneity of tumor microenvironments. Cancer cells are susceptible to genetic mutations and epigenetic alterations, which are also caused by physiochemical forces from their surroundings. This heterogeneity is important to be replicated in in vitro experiments, since it drastically affects the tumor’s response to treatment [174]. Approaches that combine 3D bioprinting and scaffold materials, together with multi-cell type 3D cultures, could support the establishment of highly complex cancer models and architectures [175]. Neal et al. successfully modeled the immune checkpoint blockade with a patient-derived organoid, which contained the original tumor T-cell population [176]. In another interesting approach, Tsai et al. developed a patient-derived organoid model including tumor, stromal and immune components of the tumor microenvironment, extremely useful for anticancer drug screening. Findings such as these may have a tremendous impact on tumor research as tumor biopsies can be directly cultured and co-cultivated with endogenous stromal cells [177].

It is widely known that cancer is a dynamic and multifactorial disease. Currently, organ-on-a-chip devices are being developed, moving research from single organ/tissue scale toward more system-like models [178]. An ideal goal of such systems would be the manufacturing of personalized platforms based on patient cells [179]. Still, several obstacles exist, including technical issues, as well as the validation of 3D modeling of in vivo tissues.

## Figures and Tables

**Figure 1 ijms-24-06949-f001:**
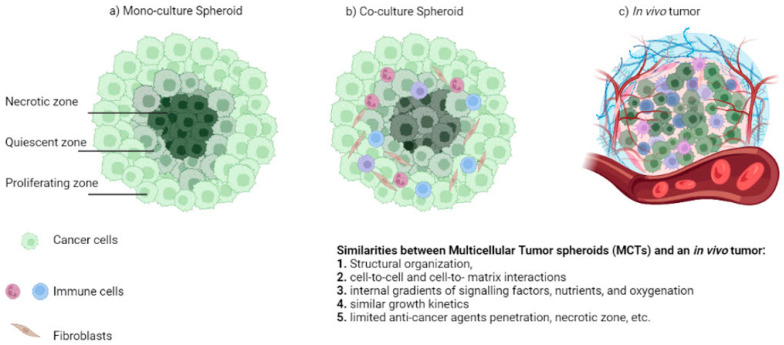
Characteristics of (**a**) mono-culture MCTS, (**b**) co-culture MCTS and (**c**) in vivo tumor.

**Figure 2 ijms-24-06949-f002:**
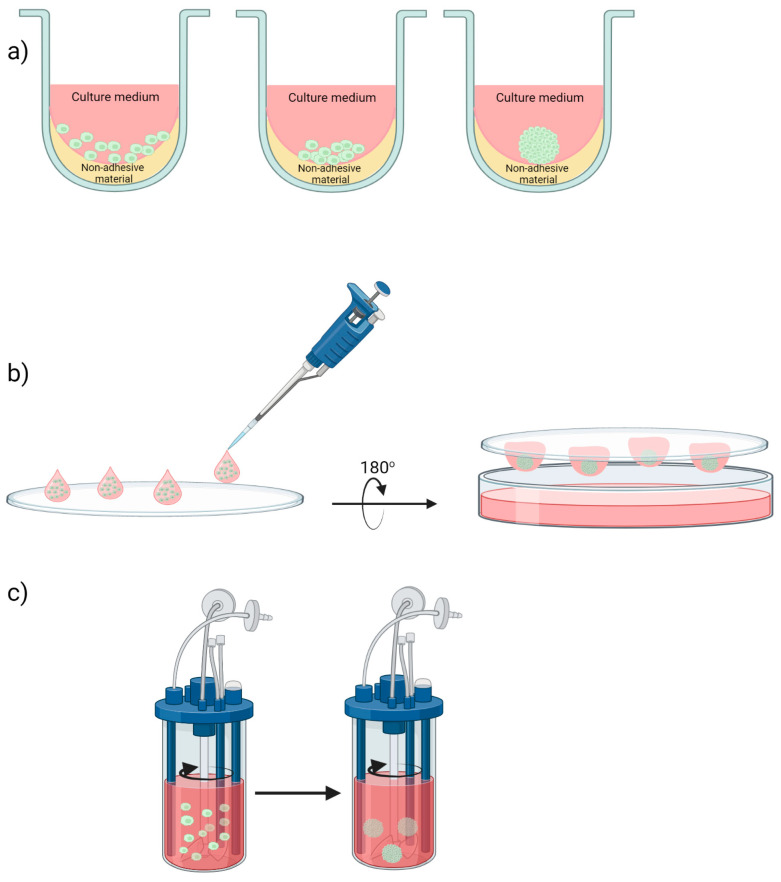
Different scaffold-free methods for MCTS formation. (**a**) Liquid Overlay Technique, (**b**) Hanging drop assay and (**c**) Spinner cultures.

**Figure 3 ijms-24-06949-f003:**
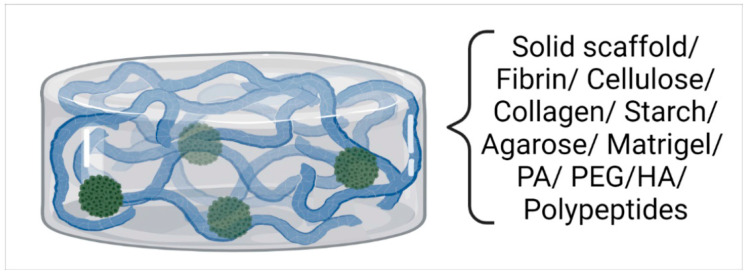
Scaffold-based methods for MCTS formation.

**Figure 4 ijms-24-06949-f004:**
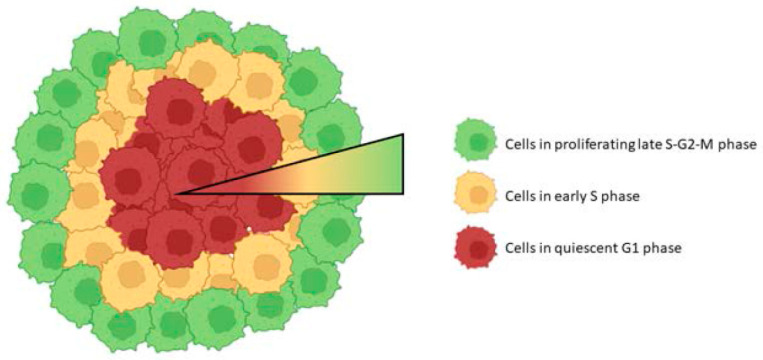
The alteration of the cell cycle stages from outer to inner layers of MCTS.

**Figure 5 ijms-24-06949-f005:**
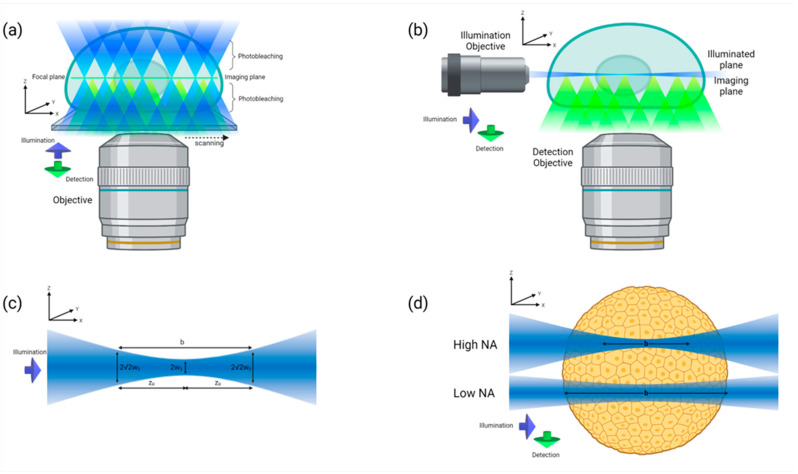
(**a**) In a confocal system the same objective is used to excite fluorophores and detect the emitted fluorescence. As the sample is scanned along the imaging plane, fluorophores bellow and above the imaging plane are also getting excited (blue cones of light). Since a pinhole is used to achieve optical sectioning, only photons emitted from the imaging plane (green cones of light) are finally detected by the photo-sensor; thus, unnecessary photobleaching and photodamage occurs. In case of a z-scan, each point of the sample is illuminated multiple times. (**b**) In a light-sheet based system, a thin “sheet” of light (blue light beam) is used to illuminate all fluorophores on the imaging plane simultaneously. Fluorescence from that plane is collected with an objective lens that is positioned orthogonally to the illumination axis and is detected using an array detector. In a case of z-scan, each plane is illuminated once, minimizing photobleaching and photodamage. (**c**) Profile of the illumination beam. The axial resolution in a light-sheet microscope is affected by the thickness of the illumination beam which is measured at the beam “waist”; the point where the beam width is smaller (denoted as 2w_0). The distance from the beam waist to the points where the beam width is equal to 2√2 w_0 is called Rayleigh range (z_R). The confocal parameter (b) is defined as the distance where the beam width is nearly homogeneous and is calculated as b=2z_R. The confocal parameter defines the effective FOV. (**d**) Beam profiles of light-sheets created with high or low numerical apertures (NA). Light-sheets created with high NA have thin beam waist and can be used to increase the axial resolution. However, the length in which the beam is homogeneous (b; confocal parameter) is short and the plane is illuminated heterogeneously. On the other hand, light-sheets created with low NA have wider beam waist and achieve lower axial resolution. However, their confocal parameter (b) is long enough to illuminate the whole plane homoge-neously. The thickness of a light-sheet should be selected that even illumination is achieved over the plane.

**Table 1 ijms-24-06949-t001:** Summary of the advantages and disadvantages of the scaffold-free systems for MCTS formation.

3D System	Advantages	Disadvantages
Liquid overlay cultures [29]	High reproducibility Simple Cost-effective No equipment required Various sizes Single MCTS monitoring Small amount of drug required Commercially available plates may “replace” laborious coating Co-cultures	For up to medium-throughput experiments Lack of cell–matrix interaction
Hanging drop assays [30]	High reproducibility Cost-effective No equipment required Size control Single MCTS monitoring Can be adapted to high-throughput screening devices	Limited sizes Lack of cell–matrix interactions Difficulties in manipulation and medium renewal
Spinner flasks [28,34]	Large-scale spheroid formation Culture nutrient enhancement Waste disposal Homogeneity of the culture medium	Expensive instruments required Heterogeneous MCTS populations Large quantities of culture medium used Not ideal for drug testing Lack of cell–matrix interactions Mechanical damage of cells
Rotational culture systems [36]	Large-scale spheroid formation Culture nutrient enhancement Waste disposal Homogeneity of the culture medium	Expensive instruments required Heterogeneous MCTS populations Large quantities of culture medium used Not ideal for drug testing Lack of cell–matrix interactions

## Data Availability

No new data were created, or where data is unavailable due to privacy or ethical restrictions.

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
