# Peer review of "Applications and Advances of Multicellular Tumor Spheroids: Challenges in Their Development and Analysis"

_ijms, 2023, doi:10.3390/ijms24086949_

Round 1
Reviewer 1 Report
This interesting review aimed to analyze recent advances in 3D cell cultures,specifically in spheroids, and the new methods that exist for their analysis.
The article is of general interest to cell biologists and is well written and up to date.
I only have some commentaries
1.- Since the review focuses on spheroids and does not mention other types of 3D cultures such as organoids or explants, I suggest keeping the title specific; for example: the phrase "development and analysis of spheroids" can be added to the title2.- In figure 1A indicate the zones mentioned on page 1 second paragraph, line 4: “Due to the nutrient and oxygen gradient, different zones are defined inside the spheroid, accompanied by differential cellular behavior (Figure 1A)”
3.- Some representative references should be added to the tables, to facilitate a quick reference to the reader.4.-In page 6, second paragraph, define 5FU
5.- Throughout the entire text, homogenize the use of the word “Spheroid” or the use of the abbreviation “MCTS”.Author Response
Firstly, we would like to thank you for your valuable comments and interesting recommendations.
Our responses to your comments are attached below.
Reviewer 1
This interesting review aimed to analyze recent advances in 3D cell cultures,
specifically in spheroids, and the new methods that exist for their analysis.
The article is of general interest to cell biologists and is well written and up to date.
I only have some commentaries
1.- Since the review focuses on spheroids and does not mention other types of 3D cultures such as organoids or explants, I suggest keeping the title specific; for example: the phrase "development and analysis of spheroids" can be added to the title
Our response: Thank you for your comment! We have updated the manuscript title to “Applications and advances of Multicellular Tumor Spheroids: challenges in their development and analysis”, and your suggestion has been incorporated therein.
2.- In figure 1A indicate the zones mentioned on page 1 second paragraph, line 4: “Due to the nutrient and oxygen gradient, different zones are defined inside the spheroid, accompanied by differential cellular behavior (Figure 1A)”
Our response: Thank you for your recommendation. We have indicated the different zones of a spheroid in figure 1A.
3.- Some representative references should be added to the tables, to facilitate a quick reference to the reader.
Our response: Representative references are added in Tables 1 and 2, as requested.
4.-In page 6, second paragraph, define 5FU.
Our response: We have now defined 5FU as 5-Fluorouracil.
5.- Throughout the entire text, homogenize the use of the word “Spheroid” or the use of the abbreviation “MCTS”.
Our response: Thank you for your comment. We now use the abbreviation “MCTS” throughout the text.

Reviewer 2 Report
In this review work, the authors discussed the advantages and disadvantages of several up-to-date methods used to construct MCTS and we present advanced methods for analyzing MCTS features. As MCTS more closely mimic the in vivo tumor environment, compared to 2D monolayers, they can evolve to be an appealing model for in vitro tumor biology studies.
There are some concerns which needs to be addressed:
==> Title does not reflect the manuscript layout. I feel somewhere spheroid could be incorporated.
1. In introduction, the first few paragraphs must address the complete basics of the model.
2. There are several places where the fonts are complete different from each other.
3. I feel, there is a need of complete manuscript update looks like lack of the subject background. I do not agree from the title manuscript layout/organization.
Author Response
Firstly, we would like to thank you for your valuable comments and interesting recommendations.
Our responses to your comments are attached below.
==> Title does not reflect the manuscript layout. I feel somewhere spheroid could be incorporated.
Our response: Thank you for your valuable comment. We have now altered the Title manuscript so as to better reflect its content. Thus, the new Title is “Applications and advances of Multicellular Tumor Spheroids: challenges in their development and analysis”.
- In introduction, the first few paragraphs must address the complete basics of the model.
Our response: We have rearranged the introduction part so as to better reflect the structure and features of the MCTS model.
- There are several places where the fonts are complete different from each other.
Our response: Thank you for your remark. We have re-checked the whole manuscript and corrected the different fonts.
- I feel, there is a need of complete manuscript update looks like lack of the subject background. I do not agree from the title manuscript layout/organization.
Our response: We thank you for your critical point. As mentioned above, we have changed the manuscript title, we re-arranged the introduction section and we also embedded updated information.

Reviewer 3 Report
This review is interesting. One comment.
2.2 Scaffold-based systems
In the current section, it is impossible to understand why each biomaterial is suitable for 3D culture. The authors should introduce the mechanisms based on the components or structure.
The authors can add appropriate references to be cited.
Cancers 2020, 12(10), 2754
Tissue Engineering Part B: Reviews.Jun 2010.351-359.http://doi.org/10.1089/ten.teb.2009.0676
Author Response
Firstly, we would like to thank you for your valuable comments and interesting recommendations.
Our responses to your comments are attached below.
This review is interesting. One comment.
2.2 Scaffold-based systems
In the current section, it is impossible to understand why each biomaterial is suitable for 3D culture. The authors should introduce the mechanisms based on the components or structure.
The authors can add appropriate references to be cited.
Cancers 2020, 12(10), 2754
Tissue Engineering Part B: Reviews.Jun 2010.351-359.http://doi.org/10.1089/ten.teb.2009.0676
Our response: Thank you for your recommendation. Information related to the characteristics of each scaffold were added in our review. Moreover, the recommended references and other appropriate references were included in our manuscript.
